# Chronic Endothelial Dysfunction after COVID-19 Infection Shown by Transcranial Color-Coded Doppler: A Cross-Sectional Study

**DOI:** 10.3390/biomedicines10102550

**Published:** 2022-10-13

**Authors:** Marino Marcic, Ljiljana Marcic, Sanja Lovric Kojundzic, Maja Marinovic Guic, Barbara Marcic, Kresimir Caljkusic

**Affiliations:** 1Department of Neurology, University Hospital Center Split, Spinciceva 1, 21000 Split, Croatia; 2Department of Radiology, Polyclinic Medikol, Soltanska 1, 21000 Split, Croatia; 3University Department of Health Studies, University of Split, Rudera Boskovica 35, 21000 Split, Croatia; 4Department of Radiology, University Hospital Center Split, Spinciceva 1, 21000 Split, Croatia; 5Department of Medical Genetics, School of Medicine, University of Mostar, 88000 Mostar, Bosnia and Herzegovina

**Keywords:** endothelial dysfunction, post-acute COVID-19, vasoreactivity, transcranial color-coded Doppler

## Abstract

In addition to respiratory symptoms, COVID-19 often causes damage to many other organs, especially in severe forms of the disease. Long-term consequences after COVID-19 are common and often have neurological symptoms. Cerebral vasoreactivity may be impaired after acute COVID-19 and in our study, we wanted to show how constant and reversible are the changes in brain vasoreactivity after infection. This cross-sectional observational study included 49 patients diagnosed with COVID-19 and mild neurological symptoms 300 days after the onset of the disease. We used a transcranial color-coded Doppler (TCCD) and a breath-holding test (BHT) to examine cerebral vasoreactivity and brain endothelial function. We analyzed the parameters of the flow rate through the middle cerebral artery (MCA): peak systolic velocity (PSV), end-diastolic velocity (EDV), mean velocity (MV), resistance index (RI) and pulsatility index (PI), and we calculated the breath-holding index (BHI). Subjects after COVID-19 infection had lower measured velocity parameters through MCA at rest period and after BHT, lower relative increases of flow velocities after BHT, and lower BHI. We showed that subjects, 300 days after COVID-19, still have impaired cerebral vasoreactivity measured by TCCD and they have chronic endothelial dysfunction.

## 1. Introduction

Following the start of the year 2020, COVID-19 spread around the globe in just a few months. The COVID-19 pandemic after more than two years has resulted in more than 500 million infected patients worldwide and more than 6 million deaths from infections [1]. COVID-19 infection is a multisystem disease and chronic complications often develop after the acute phase of the disease [2]. Elderly patients, with many comorbidities or immunocompromised patients, are more likely to develop more severe acute forms of the disease or multisystem disease, but also post-acute complications of COVID-19 infection [3]. Chronic lung disease, diabetes, obesity, cardiovascular disease, hypertension, and age over 65 years are risk factors for post-acute complications caused by COVID-19 infection [4].

The SARS-CoV-2 virus belongs to the group of RNA viruses [5], that binds to the ACE-2 receptor in the host cell via protein S [6]. ACE-2 receptors are found on the surface of many cells of the human body, their number increases with age and increased expression is associated with more severe forms of COVID-19 infection [7]. An important part of the pathophysiology of COVID-19 infection is the cytokine storm, which is related to the excessive production of inflammatory cytokines and loss of negative feedback to the immune system [8]. Cytokine storm is a life-threatening condition that often requires treatment in intensive care units and causes high mortality among patients. It is the underlying cause of acute respiratory distress syndrome (ARDS), systemic inflammatory response, and multiorgan failure in patients with COVID-19 infection [9], but it is also responsible for the long-term complications of the COVID-19 infection, including neurological symptoms [10].

The neurological manifestations of COVID-19 are a consequence of nonspecific complications of infectious diseases, direct inflammation of nerve tissue, and the inflammatory process of neuroendothelial cells [11]. All this can result in endothelial dysfunction [12], complement-induced coagulopathy [13], and vascular lesions on brain MRI scans (T2 and FLAIR) which are manifested as white matter hyperintensities [14].

The spectrum of COVID-19 symptoms can vary depending on the severity of the infection [15]. Neurological symptoms can also occur in mild forms of the disease, more severe clinical pictures are associated with more severe forms of infection, especially in hospitalized patients [16]. The COVID-19 infection can affect both the central [17] and peripheral nervous systems [18] and even lead to relapse and some rare syndromes such as Kleine–Levin syndrome [19]. There are few randomized studies that have addressed post-acute COVID-19 neurological symptoms. The most common neurological symptoms that persist after the onset of COVID-19 infection are headache, loss of smell and taste, memory and cognitive disturbances, malaise, fatigue, and insomnia, which can last up to 9 months from the start of COVID-19 [20].

In our study, we again used transcranial color-coded Doppler sonography (TCCD) and a breath-holding test (BHT) to assess brain endothelial function in a state of induced hypercapnia. After COVID-19, long-term damage to target organs can be expected in some patients, post-acute COVID-19 complications may be present after a mild form of the disease, and very often have neurological symptoms. This study aimed to determine whether changes in cerebral vasoreactivity after COVID-19 are long-lasting and whether there are still signs of endothelial dysfunction in the post-acute phase of infection, which could be a serious sequel of COVID-19 infection.

## 2. Materials and Methods

### 2.1. Study Design

In this study, we analyzed data from the electronic database of the University Hospital Center Split and found patients who sought help because of nonspecific neurological symptoms after COVID-19 infection from January to March 2021. The Ethics Committee of the University Hospital Split approved this study in March 2021 (class 500-03/21-01/39, NO 2181-147-01/06M.S-20-02, March 2021). From each participant, we obtained written informed consent. We conduct a cross-sectional observational study. We found 49 subjects who met our inclusion criteria and without any exclusion criterion (Table 1). The subject included in the study had a mild picture of COVID-19 infection according to WHO criteria [21], they were first tested in the acute phase of COVID-19 40 days after the onset of infection, and the second phase of TCCD testing was 300 days after the onset of COVID-19 infection. We excluded all subjects without an adequate temporal bone insonation window, all subjects under 30 years of age due to increased vascular elasticity, and all those who had significant risk factors for cerebrovascular disease (Table 1). Subjects for the control group were recruited from healthcare professionals and postgraduate students. Subjects in the control group had not had COVID-19 infection in the past and had a negative real-time reverse PCR test and negative nasal/pharyngeal swab test for SARS-CoV-2.

### 2.2. Sample Size and Cohort

Since our testing included a control group, it was necessary to determine the optimal sample size. The analysis was based on the main outcome of the study, which was the change in flow velocities through the MCA (PSV, EDV, MV, RI, PI) after the BHT, and the arithmetic means of these variables. We assessed the variability of the parameters in a pilot study, and, based on the pilot study, in order to prove a statistically significant difference in the arithmetic means of changes in flow rates by *t*-test for independent variables, for the group of subjects after COVID-19 infection and the control group, we needed a sample of 48 subjects per group in terms of MV values, with *p* < 0.05 and for study strength of 90%. For PSV, EDV, RI, and PI values, to also prove a significant difference in the arithmetic mean of changes in the velocity *t*-test for independent variables, with *p* < 0.05 and study power of 90%, we needed a sample size of 45 subjects in each group, so based on these calculations, we decided to include 49 subjects in each group.

### 2.3. Clinical Evaluation and TCCD Measurements

During January 2022, we collected all demographic data (data were collected by two independent neuroradiologists, and neurological examinations and TCCD measurements were performed by two neurologists). All tests were conducted on a Hitachi Aloca Arrieta 70 (2.0 MHz transducer) ultrasound machine. The study included measurements during the rest period with normal breathing and after the breath-holding test (BHT). Measurements were repeated three times for each subject. Cerebral blood flow velocity signals were monitored using a 2.0 MHz pulse probe at a depth of 52–64 mm through the right temporal bone insonation window. We insonated the right anterior cerebral artery (ACA), middle cerebral artery (MCA), and posterior cerebral artery (PCA), but all measurements were performed on the MCA. Measurements in our study included peak systolic velocity (PSV), end-diastolic velocity (EDV) and mean velocity (MV), resistance index (RI), and pulsatility index (PI) of the right MCA. We continuously measured PSV, EDV, MV, RI, and PI values on the right MCA during rest periods and after BHT, and for each measurement, we made an insonation angle correction to obtain the most accurate velocity parameters.

### 2.4. Statistics

Data were analyzed with the SPSS 20th statistical program. Statistical significance was set at *p* < 0.05, and all confidence intervals were given at 95%. For numerical variables, the Shapiro–Wilk test was used to indicate deviation from the normal distribution. Numeric variables are shown by mean (Q1–Q3) or mean ± SD. The statistical significance of the differences in categorical variables was calculated from the χ^2^ test. Analysis of the differences in numerical variables between the two groups was performed by *t*-test for independent samples or Mann–Whitney U test. The analysis of the differences between the two measurements of the numerical variables was performed by the *t*-test of paired samples. We used the Wilcoxon signed-rank test to compare the differences in repeated measurements. Analysis of differences in numeric variables between groups was performed by the Kruskal–Wallis test.

## 3. Results

### 3.1. Participants

In total, we recruited 150 subjects in the COVID-19 group and 150 subjects in the control group. In the COVID-19 group, 135 of them had a mild form of infection, and only 106 of them were in the age group between 30 and 60 years. Of the total number of 106 subjects, 84 of them meet the criteria that the COVID-19 infection started between 290 and 310 days ago. Another 20 subjects were excluded from the study because of taking various medications, either for COVID-19 infection or for other diseases. Another 15 subjects were excluded from the study because they had significant risk factors for cerebrovascular disease. In the control group, the main reasons for exclusion from the study, were an inappropriate age group, significant comorbidity, and the use of various medications. A total of 99 subjects were included in our study; 49 subjects had confirmed COVID-19 infection and 50 controls. Groups were matched by age, gender, body mass index, risk factors for cerebrovascular disease, and harmful habits such as smoking and alcohol consumption. Figure 1 shows the study flowchart.

Table 2 shows demographic data and some of the clinical parameters of these subjects. The median age of the COVID-19 group was 47 ± 6, the group included 49% females and had a body mass index (BMI) of 26 ± 3.3.

In the control group, 36 (72%) received the COVID-19 infection vaccine, 20 (55%) received the BioNTech (Mainz, Germany) vaccine, 9 (25%) received the Johnson & Johnson (New Brunswick, NJ, USA) vaccine, and 7 (20%) received the Moderna (Cambridge, MA, USA) vaccine. Among the subjects who had COVID-19 infection, 23 (47%) received additional vaccinations, 15 (65%) received the BioNTech vaccine, 5 (22%) received Johnson & Johnson vaccine, and 3 (13%) received the Moderna vaccine. Table 3 shows the vaccination status against COVID-19 infection in the group of subjects who had COVID-19 and in the control group.

Although 300 days passed after the onset of COVID-19 infection, some subjects still had some neurological symptoms. The most common symptoms were dizziness (10%) and headache (14%), but two subjects (4%) showed symptoms of cognitive impairment and one subject (2%) had a sleep disorder.

Table 4 shows the neurological symptoms that subjects had 300 days after the onset of COVID-19 infection.

Table 5 displays some laboratory values of a control group and a COVID-19 group 300 days after infection. There was no statistically significant difference in laboratory values between the two study groups.

### 3.2. Comparison of Flow Velocities through Middle Cerebral Artery at Rest and after Breath Holding Test between Test Groups

We first examined the flow rates through the MCA at rest period and after BHT. Subjects after COVID-19 infection had statistically significantly lower all measured velocities parameters through MCA at rest period and after BHT compared to the control group. The mean value of breath holding time in both groups was 38 ± 3 s (minimum 30 to maximum 45 s), and there was no statistically significant difference between the two groups of subjects (T = 0.961; *p* = 0.275). Table 6 represents flow velocities through MCA at the rest period and after BHT for groups of subjects 300 days after COVID-19 infection and the control group.

After the breath-holding test (BHT), in the group of subjects who had COVID-19, there was an increase in flow rates (PSV, EDV, MV) compared to the resting phase, but this increase was statistically significantly lower than in the control group of subjects. The resistance index (RI) and pulsatility index (PI) increased in the COVID-19 group, while in the control group they had lower values after the BHT (Table 7).

In relation to these parameters, we calculated the breath-holding index (BHI) as the percentage of increase in velocity parameters divided by breath-holding time (BHT). BHI was statistically significantly lower in subjects who had COVID-19 infection than in the control group (Table 7).

### 3.3. Comparison of the Flow Rate through the MCA at Rest and after BHT in the COVID-19 Group 40 Days and 300 Days after COVID-19 Infection

In the next part of the study, we tried to compare the flow rate parameters for the group of subjects who had COVID-19, data we collected 40 and 300 days after the start of the infection. We separately examined the flow parameters at rest period and after BHT. Table 8 shows the arithmetic means of the MCA flow rate ± SD (95% CI) at rest period for 40 days and 300 days from the onset of COVID-19.

At rest period, examined velocity parameters through MCA (PSV, EDV, MV, RI, PI) did not change statistically significantly 300 days after the onset of COVID-19 compared to values 40 days after the onset of COVID-19.

Table 9 shows the arithmetic means of the MCA flow rate ± SD (95% CI) after the BHT 40 days and 300 days after the onset of COVID-19 infection.

After the BHT, the examined velocity parameters through MCA (PSV, EDV, MV, RI, PI) did not change statistically significantly 300 days after the onset of COVID-19 compared to values 40 days after the onset of COVID-19.

### 3.4. Comparison of Relative Changes in the Examined Flow Rates through the MCA in Relation to the Time Elapsed since the Onset of COVID-19 Infection

Table 10 shows the median (IQR) changes in flow rates through the MCA relative to the time elapsed since the onset of COVID-19 infection in subjects.

There was no statistically significant difference in changes of relative values for all measured velocites parameters [PSV (Z = 1.9; *p* = 0.211), EDV (Z = 2.6; *p* = 0.037), MV (Z = 2.9; *p* = 0.015), RI (Z = 2.1; *p*= 0.691) and PI (Z = 1.26; *p* = 0.199)].

## 4. Discussion

In our previous work, we showed that patients after even mild COVID-19 infection and nonspecific neurological symptoms have impaired cerebral vasoreactivity. Impaired cerebral vasoreactivity manifests as a decreased BHI [22]. Subjects in this study were tested 300 days after the onset of COVID-19 infection. Our goal was to check the state of cerebral vasoreactivity after an extended period of time, and whether the flow parameters returned to normal values. We tested a total of 49 subjects in the group after COVID-19 and 50 in the control group. We used TCCD and BHT, we measured the same flow rate parameters for each subject (PSV, EDV, MV, PI, RI). We insonated the study groups on the same ultrasonic device, under the same circumstances (insonation of the right MCA, correction of the angle of insonation for every subject, three tests at rest period, and three times after BHT).

In subjects, we tested 300 days after the onset of COVID-19, the dominant neurological symptoms which lasted were headaches. Only two subjects still had problems with taste, and five of them had vertigo. Two subjects developed cognitive impairment and were referred for further treatment by a neuropsychologist. One subject had sleep problems that he did not have before COVID-19.

There were no statistically significant differences in laboratory values between the two examined groups.

When we compared the two groups, after the rest period and after BHT, all tested flow values were statistically significantly lower in the group after COVID-19. An increase in flow velocities that must occur physiologically after the BHT is lower in the group of subjects after COVID-19 than in the control group. Consequently, the BHI in the COVID-19 group is still lower than in the control group, 300 days after the onset of the disease.

We compared the flow velocity parameters through the MCA for the COVID-19 group, 40 and 300 days after the onset of infection. In the resting phase and after BHT, all measured parameters 300 days after the onset of COVID-19 did not show a statistically significant improvement compared to measurement 40 days after the onset of the disease. All this suggests that vasoreactivity parameters have not improved statistically significantly over s control period of 300 days. The SARS-CoV-2 virus changed brain vasoreactivity over a longer period of 300 days, and these data directly indicate impaired vasoreactivity and a poor vasodilatory response to hypercapnia induced by the BHT.

Impairment of cerebral vasoreactivity in patients after COVID-19 may be due to dysregulation of the renin-angiotensin system (RAS) as a result of the action of a SARS-CoV-2 virus on ACE-2 receptors [23], but also as a direct consequence of endothelial cell infection and diffuse endothelial inflammation. This results in endothelial dysfunction and impaired microcirculatory function, in some cases with the formation of microthrombus [24]. Our study pointed out the possibility that COVID-19 infection, regardless of its severity, could lead to cerebral vasoreactivity disorders. Due to impaired cerebral hemodynamics, these patients may be at increased risk for developing a cerebrovascular disease in the future. None of our subjects had risk factors for brain vascular disease. With the BHT, we caused hypercapnia and increased blood flow through the blood vessels [25]. The poor response to hypercapnia after a BHT in COVID-19 patients is due to the inability of the intracranial blood vessels to respond to more demanding metabolic needs [26].

Other, recently published research suggests that some patients with COVID-19 may develop post-acute respiratory, neurological, cardiac, or renal symptoms. Carfì et al. showed that 87.4% of patients who had recovered from COVID-19 have at least one symptom left behind after infection. The most common symptoms after the acute phase were fatigue and dyspnea, anosmia was present in 18% of them, and headache and dysgeusia in 10% of subjects who had COVID-19 infection [27].

Nuzzo D. et al. showed that as more and more people are infected in the world, there are more and more neurological symptoms, but also neurological complications that occur after the acute phase of COVID-19 [28]. Post-COVID-19 neurological sequelae may persist over a long time after acute infection. They may be a direct consequence of viral infection (invasion of brain tissue) or may be due to hypoxia that accompanies a severe respiratory disease. Cerebrovascular disease could be a consequence of endothelial dysfunction. Strong host immune response to COVID-19 infection, but also psychosocial pathology, can lead to long-term neurological symptoms. A small proportion of post-COVID-19 patients may have impaired sleep, memory, concentration, and attention for more than one year after acute infection. Nuzzo D. et al. concluded that the rehabilitation of patients who had COVID-19 must be adapted to each patient individually, and attention should be paid to all organs that can be the target of the virus.

A recent follow-up study by Carvalho-Schneider C. et al. found that two-thirds of patients with a milder form of COVID-19 still had some symptoms 60 days after the onset of infection, and one-third were still ill or in a demanding clinical situation [29]. Symptoms that occurred after acute COVID-19 and which lasted, occurred in all age groups, from 40 to 70 years of age. Not only does acute COVID-19 require careful medical monitoring, but also a post-acute COVID-19 sequel requires a clinician’s attention a long time after the end of the acute phase of the disease and that applies to all age groups.

For patients who had more severe clinical symptoms and who were treated in the hospital due to COVID-19 disease, a large number of them had some symptoms even 110 days after discharge from the hospital. Garrigues E. et al. showed that between their symptoms fatigue and shortness of breath dominate, but neurological symptoms such as memory loss (34%) and sleep disturbance (30.8%), and concentration disorder (28%) were also common [30]. In all these patients, quality of life was reduced (mobility, self-help, pain, anxiety or depression, normal activity), and these data showed again the need for long-term and multidisciplinary monitoring of COVID-19 patients. Post-acute COVID-19 syndrome can occur in a large number of patients who have recovered from the acute infection. According to Yan et al., the post-acute COVID-19 symptoms may be similar to the acute phase of the disease, may last a long time after the acute infection, and may affect numerous organs [31]. The most vulnerable part of the brain to endothelial dysfunction, hypoxic injury, and microbleeding is the brain stem, which contains the vital centers for cardiovascular, pulmonary, and neurological function. According to Yan et al., the most common neurological symptoms of post-acute COVID-19 are mood changes, headaches, memory disorders, dizziness, fatigue, and attention deficit disorder.

Some post-COVID-19 patients had a diminished reactive hyperemia index (RHI) which indicates endothelial dysfunction [32]. Haffke et al. used peripheral arterial tonometry (PAT) in patients after COVID-19 who had myalgia, encephalomyelitis, and chronic fatigue syndrome and compared markers of endothelial dysfunction such as Endothelin-1 (ET-1), Angiopoietin-2 (Ang-2), Endocan (ESM-1), intreleukon-8 (IL-8) and Angiotensin-Converting Enzyme (ACE). Elevated ET-1 levels were found in post-COVID-19 patients only, indicating that endothelial hypoperfusion plays a role in COVID-19 patients and can provide a rationale for future therapy. Endothelial dysfunction plays a key role in the pathogenesis of COVID-19 infection, both acute disease (and its complications) and post-acute symptoms. Ambrosino et al. investigated brachial artery flow-mediated dilation (FMD) and concluded that the post-acute syndrome of COVID-19 is associated with endothelial dysfunction, which is proportional to the severity of the lung disease, but also more pronounced in males [33].

COVID-19 patients, especially with severe respiratory syndrome, have been associated with high stroke risk. According to Eklind et al., stroke risk among hospitalized patients was 0.5–3%, and in critically ill patients was 6% [34]. Strokes of different subgroups occurred in these patients, such as small vessel disease, cardioembolic strokes, as well as cerebral venous thrombosis, and parenchymal hemorrhages. This heterogeneity suggests that the mechanisms of a stroke may not be specific to a particular pathophysiological feature, but rather the result of thrombophilia, endothelial dysfunction, thrombotic microangiopathy, and nonspecific effects of inflammation. Our study shows how persistent endothelial dysfunction is after COVID-19, and how important it is to examine this function even in milder forms of the disease.

COVID-19 patients with ischemic stroke were compared with control stroke patients and McAlpine et al. found that COVID-19 patients had lower rates of tissue-type plasminogen activator (tPA) administration, higher rates of large vessel occlusions, and higher levels of acute phase reactants and cytokines [35]. The authors noticed a strong correlation between high levels of interleukin-6 (IL-6) and soluble interleukin-2 receptor (sIL-2R) and embolic stroke of undetermined source (ESUS) suggesting that the cause for embolic strokes was hypercoagulability associated with inflammation. The authors found that some patients after COVID-19 infection had persistently elevated von Willebrand factor (VWF), factor VIII, and acute phase reactants, suggesting endotheliopathy may be longstanding after infection. These findings support the theory that COVID-19 triggers a systemic inflammatory response, which results in endothelial damage, hypercoagulability, and significantly increases the risk for thrombosis. It is particularly significant that in our study the subjects did not have elevated values of acute phase reactants, yet they showed impairment of endothelial function on the TCCD testing. TCCD in combination with a breath-holding test is an excellent method of assessing cerebral vasoreactivity [36]. In neurodegenerative diseases, at least part of the pathophysiology is related to vascular injury and dysfunction of the neurovascular unit, and an increase in proinflammatory markers and expression of ACE-2 receptors can be found. The link between COVID-19 and neurodegenerative diseases can be expected. Fotuhi et al. in their study showed all the possible neurological sequels of COVID-19 and pointed to the short-term and long-term implications that may occur in these patients, especially Parkinson’s disease, depression, insomnia, cognitive disorders, Alzheimer’s disease, and concentration disorders [37].

After the COVID-19 infection, difficulties sometimes can occur in the healthiest population, such as athletes. De Sire et al. [38] examined the neuromuscular function of female volleyball players after recovering from COVID-19 and concluded that they may experience an increased risk of injury as a consequence of weak quadriceps varus/valgus stabilization during ACL stressful movements. Authors conclude that COVID-19 influenced the athletes’ body composition with weight loss, sarcopenia, and possible malnutrition. In a recent study, Callen et al. also found chronic cerebrovascular reactivity impairment in subjects after mild forms of COVID-19 infection. In their study, they used MRI that included arterial spin labeling perfusion imaging with acetazolamide stimulus to measure cerebral blood flow (CBF) and calculate cerebral vasoreactivity (CVR). The authors came to the same conclusions as we did in our study, although they used other techniques to examine cerebral vasoreactivity [39].

The small number of subjects we included in the research is the main limitation of our study. Another important limitation is the relatively short time (300 days) after which we reexamined our subjects so that the final consequences of COVID-19 on cerebral vasoreactivity are not yet clear. The TCCD technique depends a lot on the quality of the insonation window, especially on the hardness of the temporal bone of an individual subject. Additional data would be obtained if other cerebral arteries, especially anterior and posterior cerebral arteries, were insonated. Although the BHT is less effective than carbon dioxide inhalation or the application of acetazolamide in the induction of hypercapnia, for the safety of our subjects, we used it for this test. All measurements on the ultrasound machine were performed by one of the authors who have many years of experience working with TCCD, so we avoided possible interpersonal differences during the measurements. We need additional studies to assess brain vasoreactivity after COVID-19 infection, and future studies should include more patients and should be used other methods for the induction of hypercapnia. Longer longitudinal follow-ups of these patients can give us insight into the reversibility of changes caused by SARS-CoV-2 viruses. All future trials should include patients with severe clinical pictures, both respiratory and neurological.

## 5. Conclusions

In our study, we showed once again that COVID-19 infection is primarily a multisystem disease. Disorders of the target organs can be detected for a long period after an acute infection. Endothelial dysfunction may be manifested by impaired cerebral vasoreactivity, which can be accurately and safely examined by TCCD and BHT. In the future, we need longitudinal studies that will show whether changes in endothelial function after COVID-19 infection are of a permanent nature.

## Figures and Tables

**Figure 1 biomedicines-10-02550-f001:**
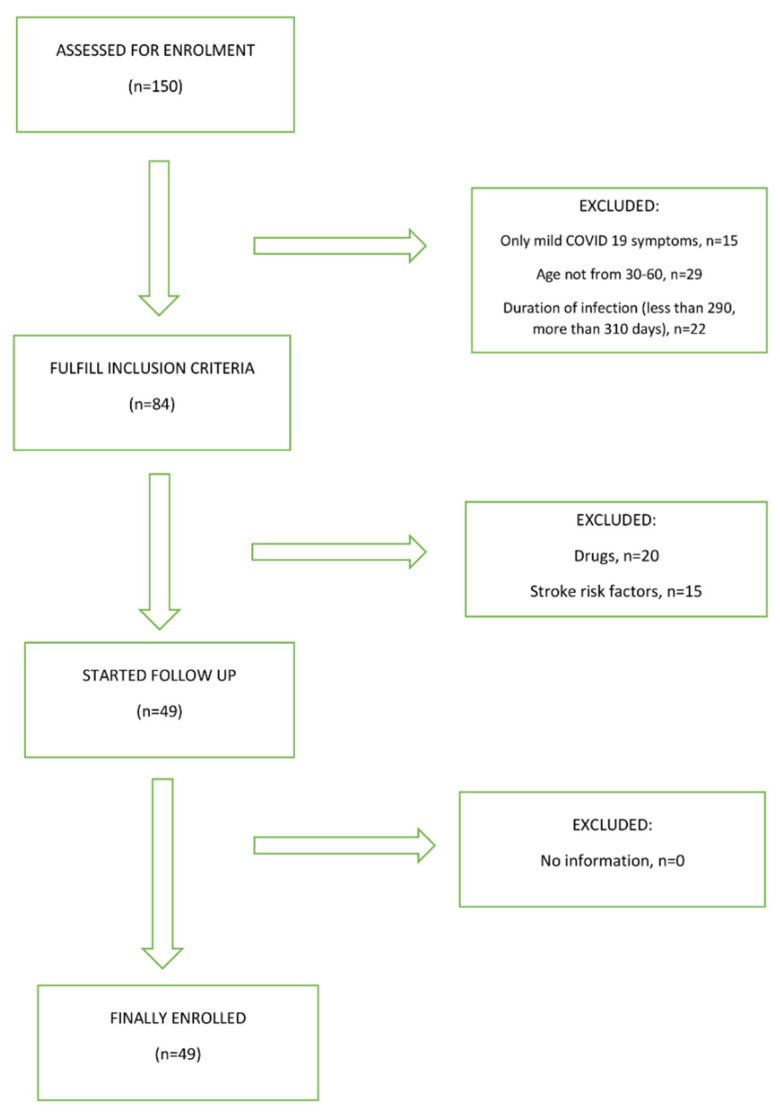
Study flowchart.

**Table 1 biomedicines-10-02550-t001:** Inclusion and exclusion criteria.

Inclusion Criteria	Exclusion Criteria
Age from 30 to 60 years	Less than 30 and more than 60 years
Mild form of COVID-19 disease	Severe or critical COVID-19 disease
Neurological symptoms: anosmia, dysgeusia, vertigo, headache, vertigo, fatigue	Severe neurological symptoms: acute cerebrovascular disease (ischemic or hemorrhagic stroke, subarachnoid hemorrhage), encephalopathy, encephalitis, meningitis, polyneuropathy, seizures, disturbance of consciousness
COVID-19 disease: from 290 to 310 days from infection start	COVID-19 disease: less than 290 or more than 310 days from infection start
Supportive therapy only	Antibiotics, corticosteroids, or oxygen use for COVID-19 infection
COVID-19 disease confirmation: positive real-time reverse PCR test, positive nasal/pharyngeal swabs test	Uncontrolled diabetes mellitus or arterial hypertension, cerebrovascular disease, hematologic disease, atrial fibrillation, chronic heart disease, cancer
	Severe alcohol consumption (more than 10 drinks per week)
	Stenosis of extracranial vertebral or carotid artery >20% or occlusive disease of intracranial cerebral arteries
	Using anticoagulant therapy, vasodilatory drugs, hormone replacement therapy, β-blocking agents, calcium channel blockers

PCR—polymerase chain reaction.

**Table 2 biomedicines-10-02550-t002:** Demographic data and some of the clinical parameters of subjects tested 300 days after the onset of COVID-19 infection.

COVID-19 Group (N = 49)			Control Group (N = 50)
Gender	M	25 (51%) *	25 (50%)
	F	24 (49%) *	25 (50%)
Diabetes melitus		5 (10%)	4 (8%)
Arterial hypertension		7 (14%)	8 (16%)
Hyperlipidemia		4 (8%)	5 (10%)
Alcohol consumption		0 (0%)	0 (0%)
Smoking		3 (6%)	3 (6%)
Age (years)		47 ± 6 **	48 ± 5 **
BMI (kg/m^2^)		26 ± 3.3 **	26 ± 3.2 **
SBP(mmHg)		125 ± 12 **	124 ± 10 **
DBP (mmHg)		77 ± 9.6 **	76 ± 9.1 **

Legend: M—male, F—female, BMI—body mass index, SBP—systolic blood pressure, DBP—diastolic blood pressure. * χ^2^ test; ** *t*-test.

**Table 3 biomedicines-10-02550-t003:** Vaccination status of subjects in COVID-19 and control group.

	COVID-19 (N = 49)	Control (N = 50)
Total vaccination (N)	23 (47%)	36 (72%)
BioNTech	15 (65%)	20 (55%)
Johnson & Johnson	5 (22%)	9 (25%)
Moderna	3 (13%)	7 (20%)

**Table 4 biomedicines-10-02550-t004:** Neurological symptoms 300 days after onset of COVID-19 infection.

Neurological Symptoms	N (49)
Anosmia	1 (2%)
Dysgeusia	2 (4%)
Vertigo	5 (10%)
Headache	7 (14%)
Fatigue	3 (6%)
Cognitive symptoms	2 (4%)
Sleep disorders	1 (2%)
Myalgia	0 (0%)

**Table 5 biomedicines-10-02550-t005:** Quantitative laboratory variables of subjects after COVID-19 infection and control group 300 days after infection.

Quantitative Laboratory Variables 300 Days after COVID-19 Infection			
	Control Group	COVID-19	*p*
Body temperature (°C)	36.5 (36–37.1)	36.5 (36–36.9)	0.791 *
Hgb g/L	153 (137–165)	149.3 (131–162)	0.249 *
Hct L/L	0.525 (0.490–0.590)	0.499(0.447–0.571)	0.067 *
Leukocytes ×10^9^/L	6.5 (4.8–8.2)	6.4 (4.9–8.1)	0.956 *
Platelets ×10^9^/L	411 (389–426)	409 (386–425)	0.121 *
Erythrocytes ×10^12^/L	5.2 (4.9–5.9)	4.9 (4.6–5.3)	0.061 *
Urea mm/L	4.9 (3.1–7.2)	5.1 (3.8–6.9)	0.557 *
Creatinine µm/L	73.5 (54–83)	75.5 (54–85)	0.138 *
Cholesterol mm/L	4.8 (4–5)	4.7 (4–5)	0.305 *
D-dimers ng/mL	0.17 (0.14–0.27)	0.19 (0.16–0.25)	0.422 *
Triglycerides mm/L	1.3 (1.1–1.6)	1.5 (1.2–1.7)	0.077 *
LDH U/L	159 (136–171)	157 (138–170)	0.097 *

Legend: Hgb—hemoglobin, Hct—hematocrit, LDH—lactic acid dehydrogenase. * Kruskal–Wallis test.

**Table 6 biomedicines-10-02550-t006:** Arithmetic means of flow velocities through MCA and standard deviations (±SD) (95% CI) of these parameters.

	Subject Groups		*p* *
	COVID-19	Controls	
At rest			
PSV (cm/s)	106 ± 12.5 (101–112)	121 ± 5.8 (117–122)	<0.001
EDV (cm/s)	52.1 ± 4.7 (50–55)	56.4 ± 5.3 (53–59)	<0.001
MV (cm/s)	72.2 ± 7.1 (68–76)	81.4 ± 7.4 (77–84)	<0.001
RI	0.52 ± 0.03 (0.50–0.53)	0.55 ± 0.03 (0.54–0.57)	<0.001
PI	0.74 ± 0.06 (0.72–0.77)	0.78 ± 0.04 (0.76–0.79)	<0.001
After BHT			
PSV (cm/s)	122 ± 11.1 (116–127)	162 ± 7.4 (157–164)	<0.001
EDV (cm/s)	69.1 ± 3.1 (66–71)	86 ± 5.6 (81–89)	<0.001
MV (cm/s)	94.3 ± 8.1 (90–98)	112.1 ± 4 (107–116)	<0.001
RI	0.53 ± 0.04 (0.51–0.53)	0.50 ± 0.01 (0.49.-0.52)	<0.001
PI	0.78 ± 0.07 (0.74–0.81)	0.75 ± 0.04 (0.73–0.77)	<0.001

Legend: PSV—peak systolic velocity, EDV—end-diastolic velocity, MV—mean flow velocity, RI—resistance index, PI—pulsatility index, BHT—breath-holding test. * *t*-test for independent samples.

**Table 7 biomedicines-10-02550-t007:** Median (Q1-Q3) presentation of changes in flow velocities rates through MCA after BHT and rest period in relation to groups of subjects.

	Subjects Groups		*p* *
	COVID-19	Controls	
Relative change of velocities parameters after BHT compared to values at resting period (%)			
∆PSV (%)	16 (11–18)	41 (37–45)	<0.001
∆EDV (%)	17 (15–24)	32 (29–37)	<0.001
∆MV (%)	22 (20–26)	33 (28–39)	<0.001
∆RI (%)	1.4 (−0.6 to 2.1)	−5 (−12 to −2)	<0.001
∆PI (%)	3.3 (1.9–3.7)	−4 (−6 to −1.4)	<0.001
BHI	0.43 (0.35–0.54)	0.97 (0.88–1.07)	<0.001

Legend: PSV—peak systolic velocity, EDV—end-diastolic velocity, MV—mean flow velocity, RI—resistance index, PI—pulsatility index, BHI—breath holding index. * Mann–Whitney U test.

**Table 8 biomedicines-10-02550-t008:** Arithmetic means of MCA flow rates ± SD (95% CI) at rest period 40 days and 300 days from the onset of COVID-19.

Flow Rates during Resting Period in a Group of Subjects after COVID-19 Infection				
	40 Days	300 Days	Difference	*p* *
PSV (cm/s)	109 ± 12 (103–115)	106 ± 12.5 (101–112)	−2.83 ± 2.4 (−2 to 0.35)	0.156
EDV (cm/s)	53 ± 4 (51–55)	52 ± 4.7 (51–54)	−1.2 ± 1.1 (−1.2 to −0.69)	0.021
MV (cm/s)	72 ± 7.4 (68–75)	72 ± 7.1 (70–76)	−0.5 ± 2.9 (−2.9 to −0.11)	0.066
RI	0.52 ± 0.027 (0.51–0.54)	0.52 ± 0.03 (0.50–0.53)	−0.01 ± 0.02 (−0.02 to 0.0008)	0.170
PI	0.75 ± 0.07 (0.73–0.79)	0.74 ± 0.06 (0.75–0.82)	−09 ± 0.02 (−1.02–−0.01)	0.128

Legend: PSV—peak systolic velocity, EDV—end-diastolic velocity, MV—mean flow velocity, RI—resistance index, PI—pulsatility index. * *t*-test for dependent samples.

**Table 9 biomedicines-10-02550-t009:** Arithmetic means of MCA flow rates ± SD (95% CI) after a BHT 40 days and 300 days from the onset of COVID-19.

	Flow Rates in MCA after BHT			
	40 Days	300 Days	Difference	*p* *
PSV (cm/s)	124 ± 10 (120–129)	122 ± 11.1 (121–131)	−1.4 ± 2.7 (−2.7 to −0.11)	0.065
EDV (cm/s)	70 ± 3.5 (68–72)	69.1 ± 3.1 (66–71)	−0.35 ± 1.1 (−0.58–0.5)	0.837
MV (cm/s)	94.1 ± 8.8 (90–98)	94.3 ± 8.1 (91–97)	0.2 ± 1.4 (−0–33–2.2)	0–118
RI	0.53 ± 0.02 (0.52–0.54)	0.53 ± 0.04 (0.52–0.55)	−0.006 ± 0.01 (−0.1–−0.0007)	0.028
PI	0.78 ± 0.08 (0.75–0.82)	0.78 ± 0.07 (0.76–0.82)	−0.008 ± 0.03 (−0.02–0.008)	0.286

Legend: PSV—peak systolic velocity, EDV—end-diastolic velocity, MV—mean flow velocity, RI—resistance index, PI—pulsatility index, BHT—breath-holding test. * *t*-test for dependent samples.

**Table 10 biomedicines-10-02550-t010:** Display of median (IQR) changes in flow rates through the MCA compared to the time elapsed since the onset of COVID-19 in subjects.

Subjects after COVID-19 Infection			
Relative Change in Value after BHT and Value at Rest (%)	40 Days	300 Days	*p* *
PSV	15(11–20)	16 (11–18)	0.211
EDV	18 (16–24)	17 (14–22)	0.037
MV	25 (21–28)	22 (20–26)	0.015
RI	1.5 (−0.5–1.9)	1.4 (−0.6–2.1)	0.691
PI	3.1 (1.7–3.5)	3.3 (1.9–3.7)	0.199

Legend: BHT—breath-holding test, PSV—peak systolic velocity, EDV—end-diastolic velocity, MV—mean flow velocity, RI—resistance index, PI—pulsatility index. * Wilcoxon signed-rank test.

## Data Availability

Not applicable.

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
