# Peer review of "Chronic Endothelial Dysfunction after COVID-19 Infection Shown by Transcranial Color-Coded Doppler: A Cross-Sectional Study"

_biomedicines, 2022, doi:10.3390/biomedicines10102550_

Round 1

Reviewer 1 Report

The authors describe an extended study as a continuation of a pilot study that the authors themselves had previously published. Overall it is well written, argued with precision and considering the literature. The reading is pleasant, the conclusions are consistent. As a suggestion for future studies, I would just like to introduce one topic: It is not entirely clear whether the cerebrovascular changes found in long covid patients are present only in symptomatic ones. That is, whether these differences are really related to the symptoms. In other words, it could be interesting to perform linear correlations between resistance, velocity or flow values with numerical clinimetric scores (mini mental score, insomnia severity index, VAS ...).

Author Response

First reviewer:

- Changes in brain vasoreactivity in our subjects after the COVID 19 infection were not related to individual symptoms that remained after COVID 19 infection. Changes in cerebral vasoreactivity were found in all subjects, and only some had residual, chronic neurological symptoms after the COVID 19 .

Cognitive impairment had two subjects, and for these two subjects with cognitive impairment, the MMS score was 22 and 23, respectively. Regarding headache, all subjects had headache severity according to the VAS scale of 3 or 4, and we did not find a correlation of pain severity with changes in flow parameters. It would certainly be interesting in future studies to see the correlation of headache severity with changes in cerebral vasoreactivity parameters, as well as the correlation of other symptoms and these velocity parameters. This will certainly require a much larger group of respondents, using more precise scales to measure pain and cognitive impairment, but that was not the scope of our research. Thanks for the helpful suggestions.

Reviewer 2 Report

Dear Authors,

In my opinion, the topic is interesting, however, I have several concerns about the methodological implant of this study, and several critical issues should be addressed.

Major revisions:

TITLE: the title is intriguing but somehow misleading given that the authors mainly focused on flow rate rather than endothelial dysfunction. The title should be improved accordingly.

METHODS: The authors should better clarify who performed the analysis (qualification, degree, and especially if blinded).

METHODS: In order to improve the quality of this paper, you should follow the STROBE guidelines for cross-sectional studies.

RESULTS: A specific ‘Participants’ subsection should be included in this section, reporting the numbers of patients assessed for eligibility, and patients excluded and clarifying at least the main cause of exclusions. A study flowchart might improve the legibility of the study results.

DISCUSSION: This section should be largely improved by highlighting the disabling sequelae of COVID-19, related to the neurological sphere and neuromuscular control.

According to this, you should cite the following references:

-       Ambrosino P, Calcaterra I, Molino A, Moretta P, Lupoli R, Spedicato GA, Papa A, Motta A, Maniscalco M, Di Minno MND. Persistent Endothelial Dysfunction in Post-Acute COVID-19 Syndrome: A Case-Control Study. Biomedicines. 2021;9(8):957. doi: 10.3390/biomedicines9080957.

-       Yan Z, Yang M, Lai CL. Long COVID-19 Syndrome: A Comprehensive Review of Its Effect on Various Organ Systems and Recommendation on Rehabilitation Plans. Biomedicines. 2021;9(8):966. doi: 10.3390/biomedicines9080966.

-       de Sire, A.; Demeco, A.; Marotta, N.; Spano, R.; Curci, C.; Farì, G.; Fortunato, F.; Ioa, T.; Lippi, L.; Paolucci, T.; et al. Neuromuscular impairment of knee stabilizer muscles in a COVID-19 cluster of female volleyball players: Which role for rehabilitation in the post-COVID-19 return-to-play?. Appl. Sci. 2022;12:557. doi: 10.3390/app12020557.

Minor revisions:

TABLES: Some Tables are not provided by legend. Please provide legends for each table.

Please use the right punctuation in decimals.

A careful English revision is necessary because some sentences are difficult to read.

Author Response

Major revision

As suggested by second reviewer, we changed the title of the paper so that we added the method by which we proved changes in cerebral vasoreactivity. So now the title of the paper is more precise as to the way in which we proved our theory. The title of the work is now „Chronic Endothelial Dysfunction After COVID 19 Infection Shown by Transcranial Color-Coded Doppler: A Cross-Sectional Study“.

As the reviewer suggested, we have indicated exactly which of the authors performed the data collection, who performed the neurological examination, and who performed the TCCD measurements.

According to the reviewer's suggestions, we adapted the structure of the Work Methods to the STROBE guidelines, as much as possible.

As suggested by the reviewer, we have included the Participants subsection at the beginning of the Results section. At the beginning of Participants subsection, we described in detail how we recruited respondents into both groups and what were the main reasons why we excluded some respondents from the study.

As it was well suggested by the second reviewer, in the Participants subdivision we have shown the image of the Study flowchart.

According to the reviewer's suggestions, we included and referred to three studies that talk about endothelial dysfunction (Ambrosino et al.), post COVID 19 syndrome (Yan et al.) and neuromuscular symptoms (de Sie et al.) In this way, we emphasized the importance of the post-acute COVID 19 syndrome, especially neurological symptoms, and warned of the possibility that the COVID 19 infection could even damage neuromuscular control.

We have also added the most recent study which also revealed impaired brain vasoreactivity after milder forms of COVID infection, and in their work they used MRI and acetazolamide.

Minor revision

As suggested by the reviewer, we provide legends for each table.

As suggested by the reviewer, we changed all for the right punctuation in decimals.

Once again, we revised the English language to make the text more understandable.

Reviewer 3 Report

The work of Marcic et al. is very interesting, and add important informations about post-COVID-19 infection. My comments are especially related to the manuscript writing which should be improved.

Minor points

Abstract: According to MDPI's guidelines abstract should be about 200 words, please reduce it and eliminate the subdivision in headings as well as the p value.

Introduction

- Please add the full form of MRI

- In general the introduction is too long for a research article, please reduce it.

Discussion

- Some sentences are repetitive like: "Haffke M. et al in a recent study found..."; "Elkind et al. found.."; "McAlpine et al. in recent study"...Please revise.

- I appreciate that authors consider the limitation of their study, this is a very important point. Thanks.

Author Response

Third reviewer:

As the third reviewer suggested, we have changed the abstract to comply with MDPI guidelines. We shortened abstract to 190 words, we eliminated the subdivisions and removed the p values. 

As the third reviewer suggested, we shortened the introduction.

As the third reviewer suggested,we add the full form of MRI sequences (T2 and FLAIR sequences).

As the third reviewer suggested, in the discussion we revised some sentences that were repeated in the same way, so that we did not start the citation of individual papers with the names of the authors.

Finally, in accordance with the Plagiarism report, we changed sentences which are highlighted in the Plagiarism report.

Round 2

Reviewer 2 Report

Dear Editor, 

the authors addressed all my concerns. But the manuscript does not fit the template of the Journal. The authors should transfer the revisions to the template so that the manuscript can be accepted. 

Best regards

Author Response

As the Academic Editor suggested, we have changed what was asked and listed point by point.
